# Understanding Disclosure Decisions and Communication About HPV-Related Cancer: A Qualitative Exploration of Stigma and Social Support

**DOI:** 10.3390/healthcare13090966

**Published:** 2025-04-22

**Authors:** Seiichi Villalona, Julian Sanchez, Preeyapat Mangkalard, Alicia L. Best

**Affiliations:** 1Department of Medicine, Perelman School of Medicine, University of Pennsylvania, Philadelphia, PA 19104, USA; 2Hospital of the University of Pennsylvania, Philadelphia, PA 19104, USA; 3Moffitt Cancer Center, Tampa, FL 33612, USA; julian.sanchez@moffitt.org; 4Morsani College of Medicine, University of South Florida, Tampa, FL 33620, USA; 5Behavioural Science and Health Research Department, Institute of Epidemiology and Health Care, University College London, London WC1E 6BT, UK; preeyapat.mangkalard.23@ucl.ac.uk; 6Department of Public Health Education, Morehouse School of Medicine, Atlanta, GA 30310, USA; abest@msm.edu

**Keywords:** human papillomavirus (HPV), cancer, disclosure, social support, HPV-related cancer

## Abstract

**Background/Objectives**: This study aimed to explore the barriers and facilitators influencing initial self-disclosure among individuals diagnosed with human papillomavirus (HPV)-related cancers, as well as examine the post-disclosure experiences of affected individuals. Emphasis was placed on understanding the roles of perceived and internalized stigma in these interpersonal communication encounters. **Methods**: Semi-structured interviews were conducted with 27 participants diagnosed with an HPV-related cancer. MAXQDA was used for qualitative analysis with themes grounded in Attribution Theory and the Disclosure Model. **Results**: Barriers to disclosure identified among participants included privacy concerns, hesitancy to burden others, and discomfort discussing the anatomic location of their cancer. In contrast, seeking support, instances of misunderstanding HPV’s relation to cancer, and the proactive detailing of their diagnosis to avoid judgment emerged as facilitators of disclosure. While many recounted positive post-disclosure experiences, some participants expressed feelings of guilt and internalized stigma, suggesting a deeper emotional struggle in communicating about their diagnosis to others in their social support networks. Nuances in the internalized stigma were observed in specific subgroups among this patient population, such as those who identify as a sexual/gender minority. **Conclusions**: This study underscores the multifaceted challenges experienced among individuals diagnosed with HPV-related cancers when disclosing their diagnoses and seeking social support. This study highlights the imperative role of identifying psychosocial distress in the post-diagnosis period among individuals with HPV-related cancer. Future research should explore ways to enhance social support for this patient population by improving healthcare providers’ screening measures and providing integrated support services earlier to better address their psychosocial needs.

## 1. Introduction

The human papillomavirus (HPV) is the most common sexually transmitted infection (STI) in the United States and is associated with a majority of cancers involving the oropharynx, anal canal, cervix, vagina, vulva, and penis [1]. As cancer mortality rates continue to decline and people are living longer after receiving a cancer diagnosis, national efforts to improve quality of life (QoL) among cancer survivors are critical [2]. Social support is essential in mitigating the negative physical and emotional effects of cancer and is one of the most important factors influencing QoL after cancer diagnosis and treatment [3,4]. Sharing a cancer diagnosis (i.e., self-disclosure) with supportive loved ones is often the first step in attaining social support. However, some individuals report hiding their cancer diagnosis and treatment, which can isolate them from various social networks and lead to psychological distress, depression, and other negative outcomes [5].

The epidemiology of HPV-related cancers has evolved over the last twenty years. In 1999, cervical cancers were the most common malignancies among the HPV-related cancers [6], with a steady decrease in incidence rates through 2015 due to effective clinical screening methods and the advent of vaccination against the aggressive strains of the virus [7]. The development of the Gardasil multivalent vaccine was initially rolled out with the focus on reaching at-risk adolescent females, which unintentionally led to the “feminization” of this STI and its primary association with cervical cancer [8]. This focus on females additionally influenced research among this group of malignancies, with a large proportion of funding and related studies centering around cervical and other gynecological cancers [8].

In contrast, the incidence rates of oropharyngeal cancers have increased in both sexes, of which males with these malignancies surpassed the rates of cervical cancers and made up the largest proportion of HPV-related cancers in 2015 [6]. Despite this shift, stigma and disclosure in HPV-related oropharyngeal cancer remain underexplored compared to cervical and anal cancers, where stigma is more recognized. The rising burden of oropharyngeal cancer highlights the need to understand its impact on disclosure and psychosocial outcomes. Although relatively rare, increases in incidence rates have additionally been observed in HPV-related anal cancers in both sexes, with females carrying a higher proportion of the disease burden relative to males, across all age groups [6]. Approximately 91% of all anal cancers are related to HPV infection [1] and are more commonly diagnosed in individuals with risk factors such as being immunocompromised (particularly with HIV), using tobacco, and having a prior history of other STIs [9]. The overall trends in the landscape of HPV-related cancers and their inherent risk factors make certain patient populations more prone to experiencing psychosocial distress upon being diagnosed with these cancers, such as men who have sex with men (MSM).

Self-disclosure is defined as the intentional communication of truthful information (that would otherwise be private) to another individual. The reasons for self-disclosure vary from person to person, and within the context of sharing a cancer diagnosis, it can serve as a means of receiving social support from individuals in one’s immediate and extended social networks. Psychologically, self-disclosure can facilitate emotional support and reduce feelings of isolation, but concerns about stigma, judgment, or strained relationships may create barriers. Disclosure is also essential for continuity of care, as open communication with support systems can improve adherence to treatment plans. Additionally, for HPV-related cancers, disclosure may involve considerations about informing partners and managing concerns about HPV transmission. One of the goals of this study was to explore barriers and facilitators to initial self-disclosure among individuals diagnosed with HPV-positive oropharyngeal, cervical, and anal cancers. Additionally, we examined participants’ experiences after disclosure, with a particular focus on the role of perceived and internalized stigma in communicating their cancer diagnosis to individuals within their support networks.

### 1.1. Interpersonal Communication and Cancer-Related Quality of Life

Self-disclosure is an integral part of social interaction and is defined as the intentional verbal or written communication of truthful information (that would otherwise be private or personal) with other individuals [10,11]. Self-disclosure can serve as a catalyst to other interpersonal communication about cancer such as expressing thoughts and feelings related to diagnosis and/or treatment [10]. The reasons for self-disclosure vary from person to person and within the context of sharing a diagnosis of cancer can serve as a way of receiving social support from individuals in one’s immediate and extended social networks. In general, talking about the cancer experience with supportive loved ones can help build personal relationships, increase social support, and positively influence quality of life (QoL) among people diagnosed with cancer [12]. On the other hand, avoiding interpersonal communication about cancer can be detrimental to mental and physical health [5]. For example, individuals diagnosed with oropharyngeal, cervical, and anal cancers have reported hiding their diagnosis because of embarrassment and/or perceived stigma as these cancers are associated with HPV, and, by default, may lead to assumptions about individual sexual behavior [13,14]. Moreover, disparities exist in cancer-related QoL by cancer type, with oropharyngeal (tonsillar, tongue, etc.), cervical, and anal cancer survivors having poorer outcomes when compared to other cancer survivors. Finally, negative responses from loved ones can further contribute to the psychosocial burden of HPV-related cancer, and these negative responses can have a greater influence on QoL than supportive responses [15,16].

#### 1.1.1. HPV-Related Cancer and Stigma

HPV-related cancers have been historically stigmatized due to their anatomic locations, which is further exacerbated by individual perceptions of stigma when considering the linkage of HPV to sexual behavior [17,18]. Additionally, promotion of the HPV vaccine has resulted in increased media coverage about the link between HPV and certain cancers, which studies have observed to contribute to stigmatization among individuals diagnosed with HPV-related cancers [19,20]. Here, stigma is defined as a “social process experienced or anticipated, characterized by exclusion, rejection, blame or devaluation” [21].

Gender and sexual orientation play a role in how individuals experience HPV-related stigma. One qualitative study found that women diagnosed with cervical cancer held internalized beliefs that others blamed them for their disease due to the assumption of sexual promiscuity [22]. Similarly, individuals with HPV-related oropharyngeal cancers reported feeling uncomfortable in sharing their diagnosis with immediate family and friends because of the condition’s link to sexual behavior [23]. Internalized stigma and the sentiments of embarrassment and shame are magnified in individuals with HPV-related anal cancers, where studies have reported this group to experience difficulties in disclosing their conditions to others within their social support networks because of the anatomic site and link to sexual behaviors [13,24,25,26].

Perceptions and/or experiences of stigma may be compounded in subpopulations of individuals diagnosed with HPV-related cancers such as MSM due to perceptions of these individuals making unhealthy choices that ultimately led to developing this anal cancer [14,27,28]. This is to say that being diagnosed with an HPV-related cancer inherently carries a “double stigma” among certain patient populations because of the direct association between perceived promiscuous sexual behavior or sexual practices (such as engaging in anoreceptive intercourse) and their sexual orientation. Lastly, because anal cancers are relatively rare, studies have reported on the isolation and psychosocial stressors these individuals experience when being diagnosed with these “orphan cancers” because of the lack of community of individuals with the same or similar diagnoses [13,26]. Here, the term “orphan cancers” refers to those that are not among the most commonly diagnosed cancers for which more communal support may be accessed among individuals with the same conditions [13]. The intersection of sexual stigma, gender, and sexual orientation in HPV-related cancers underscores the critical role of social support and social connections in mitigating psychosocial distress and improving quality of life (QoL) [28].

#### 1.1.2. Conceptual/Theoretical Framework

The present study drew from Attribution Theory and the Disclosure Model in examining interpersonal communication among individuals diagnosed with HPV-related cancers and their social networks. Attribution Theory helps to explain how individuals may perceive health outcomes by classifying causes as either personally “controllable” or “uncontrollable” [29]. This theory suggests that individuals who perceive themselves as having contributed to their health outcome through conscious behavior may be more likely to perceive being stigmatized or blamed in comparison to those whose health outcomes are believed to be the result of uncontrollable circumstances. Disparities in QoL by cancer type may be influenced by public perceptions of some cancer types resulting from “irresponsible” behavior [30], further contributing to negative psychosocial outcomes. For instance, smoking is the primary risk factor associated with lung cancer, and the literature suggests that lung cancer carries a greater burden of stigma than many other cancer types. Marlow and colleagues (2015) [31] indicated that lung cancer is often perceived as “self-inflicted,” and this public perception is largely due to media and policy which reinforces the stigmatization of lung cancer. In this study, we draw from Attribution Theory to parallel the causal behavior and perceived stigma of smoking and lung cancer to describe why HPV-related cancers may be stigmatized.

Disclosure decisions may be impacted by shame, embarrassment, and perceived stigma for certain conditions (e.g., HIV and other STIs) [13,23,24]. The Disclosure Model attempts to explain the cognitive process by which individuals make decisions about self-disclosure [32]. Ragins (2008) describes disclosure decisions as involving a balance of anticipated psychosocial benefits versus fear of stigma-related consequences. As depicted in Figure 1, the three variables considered in this model are the internal psychological processes of self-perception, anticipated consequences of disclosure, and perceived support in disclosing [32]. Taken together, Attribution Theory and the Disclosure Model provide a conceptual framework to explore the cognitive processes that individuals diagnosed with HPV-related cancers may undergo when making decisions about disclosing their illness to family or friends.

## 2. Materials and Methods

This study employed an in-depth interview methodology to explore disclosure decisions and experiences among individuals with HPV-related cancers. Eligibility criteria included the following: 18 years of age or older; diagnosed with an HPV-positive oropharyngeal, cervical, or anal cancer; undergoing active definitive treatment (with curative intent); fluent in English; and able to provide written informed consent. Individuals were ineligible from participating in this study if their primary/preferred language was not English; they had a prior medical history of cognitive impairment; or they were receiving treatment with palliative intent. In partnership with a large National Cancer Institute-designated comprehensive cancer center and associated clinical providers, participants were first identified via the hospital electronic health record system and recruited in person by a research assistant at an outpatient clinic. Participating oncology specialty clinics included Head and Neck, Gastrointestinal, and Gynecology. Each of the participating clinics experienced high patient volumes as the institution where the study was conducted served as the referral cancer center of the region. After obtaining written consent, eligible participants were interviewed in person or via telephone. Interviews ranged from 25 to 55 min, and participants received a USD 50.00 gift card for their time. A total of 27 eligible participants were recruited and consented to participate in this study. No patients were excluded during the recruitment process.

Due to the inherently personal and sensitive nature of the topics explored in this study, the research team made sure that the interviews were conducted in a private place. Prior to the formal start of the interviews, participants were reassured that any information shared would be deidentified and were empowered to only share whatever they felt comfortable discussing. The research team members practiced interviews with each other prior to the formal data collection phase of the study. Supportive listening techniques were also practiced in order to navigate emotionally charged situations during the actual interviews. All of the interviews were recorded and transcribed for formal data analysis. All data were stored in an encrypted data folder only accessible to the research team.

### 2.1. Measures

Participants provided socio-demographic information in response to a brief, verbally administered questionnaire conducted after the interview. Demographic variables included age, education level, income, race, gender, sexual orientation, marital status, type of medical care, and religious affiliation. Additionally, this questionnaire asked participants to rate their personal health and level of religiousness and spirituality.

The research team developed a semi-structured interview guide based on the literature [33,34] and previously collected data from clinicians to understand participant perceptions of the link between HPV and their cancer diagnosis. The interview guide explored the psychosocial impact of having an HPV-related cancer (perceived stigma, self-blame, etc.); the extent of interpersonal communication and perceived social support related to cancer; and the role of religion and/or spirituality in coping with cancer. Although the interview guide is published elsewhere, some example questions to elicit participant thoughts about psychosocial impacts, stigma, and religion/spirituality include the following: *Some people with *insert cancer type* cancer report problems talking about their illness with family and friends because of fear of being judged. Can you describe any of these experiences you have had? How did your family, friends, and co-workers respond to your cancer diagnosis? How do other people’s questions about/reactions to your cancer affect you? What is the role of religion and/or spirituality in coping with *insert cancer type* cancer specifically? Is there any reason you may avoid talking with your church members or pastor about your cancer diagnosis?* Data collection and analysis procedures including interview guide and codebook development are described in detail elsewhere [35].

### 2.2. Analysis Strategy

Throughout data collection, the team took an iterative approach to interpreting data and identifying themes. During this stage of the study, the research team employed the following steps: (1) using Attribution Theory and the Disclosure Model to generate a priori codes; (2) data familiarization through repeated reading of the transcribed interviews; (3) searching for themes; (4) identifying emergent themes; and (5) team-based discussions regarding the findings [36]. Attribution Theory guided the categorization of how participants perceived the causes of their diagnosis, distinguishing between internal, external, and neutral attributions. The Disclosure Model informed the coding of disclosure decisions, structuring themes around self-perception, anticipated consequences, and perceived support. This theory-driven approach ensured a systematic exploration of how individuals with HPV-related cancers navigated stigma, disclosure, and social support dynamics. A primary coder used the Applied Thematic Analysis framework [37] to identify salient and emergent themes, which were then discussed with the research team to reach consensus and data saturation. These research team meetings were used to determine when no new data were being identified, resulting in a final study sample of 27 participants. MAXQDA qualitative analysis software, version 12 [38], was used to manage qualitative data, and SPSS version 24 (IBM Corporation, Armonk, NY, USA, 2016) was used to conduct univariate analyses with quantitative data. This study was approved by the Institutional Review Board at the University of South Florida and the Scientific Review Committee at Moffitt Cancer Center.

## 3. Results

A total of 27 one-on-one in-depth interviews were conducted from September 2016 to May 2017. Table 1 depicts the demographic characteristics of the study sample. Participants in this study had a mean age of 54 years. Women and individuals reporting non-Hispanic White/Caucasian race/ethnicity represented most of the study sample. Participants diagnosed with anal cancers represented the largest proportion of study participants, followed by those diagnosed with gynecological and oropharyngeal cancers. Gynecological cancers (which included cervical, vulvar, and vaginal tumors) were grouped together because of their close anatomic proximity.

Among all the study participants, 11 reported immediately disclosing their cancer diagnoses to individuals within their immediate and extended social networks. This group consisted of four participants diagnosed with oropharyngeal cancer, five with gynecological cancer, and two with anal cancer. Most of the participants with oropharyngeal (4 of 7) and gynecological cancers (5 of 9) reported being immediately open to sharing their diagnoses, whereas only a few participants with anal cancer (2 of 11) reported feeling comfortable disclosing to others. Three overarching themes that emerged among participant responses included barriers to disclosure of cancer diagnosis, facilitators to initial disclosure, and experiences after disclosure.

### 3.1. Barriers to Disclosure of Cancer

A total of 14 participants described initial hesitancy to disclose their cancer diagnoses and reported disclosing to only a few members within their immediate social networks. Those reporting initial hesitancy to disclosure consisted of two of the seven participants diagnosed with oropharyngeal cancer, five of the nine participants diagnosed with gynecological cancers, and seven of the eleven participants diagnosed with anal cancer. Two participants diagnosed with anal cancer reported not disclosing to individuals in their social networks and exclusively discussing their cancer diagnoses with their treating oncologists. Three main reasons were described for not immediately disclosing cancer diagnoses with others: (1) cancer stigma and desire to preserve their personal privacy; (2) desire to minimize burden to others; and (3) discomfort with discussing the anatomic location of their cancer.

#### 3.1.1. Cancer Stigma and Personal Privacy

Reasons for hesitancy in initially disclosing cancer diagnoses varied across participants. A subtheme that emerged among those hesitant to disclosing their diagnoses centered on privacy and the desire to protect one’s personal information from being revealed to others.


*“And [if] people don’t ask me, I don’t, I don’t say anything. Only because, other than church and volunteer and cooking food for the homeless, I don’t have anything to do with them other than their services, the mass, and the volunteering, and that’s it. But I, I don’t talk about my personal stuff with who’s not a really close friend or family.”—0201, female diagnosed with anal cancer.*



*“I did share it with my parents at first, but then I didn’t keep them in the loop once the major surgeries were done. I didn’t keep them in the loop at all. My mother did not respect me on that, and she shared [my diagnosis] in an open venue to pray for me and she kind of broke my trust. She didn’t go through with my wishes.”—0211, female diagnosed with anal cancer.*


#### 3.1.2. Minimize Burdening Others

Another subtheme that emerged as a barrier to disclosure pertained to minimizing or mitigating the emotional burden on individuals within participants’ social networks. Some participants wanted to avoid adding additional stressors to potential caretakers such as their parents or children.


*“I wasn’t comfortable talking to my parents about it just because they’re very elderly and fragile and we just basically don’t tell them about anything that’s concerning or you know, potentially life-threatening because we don’t want to put them in a position of fear. So at the same time I was dealing with that, my brother had developed an infection from knee surgery that he had. It was kind of serious and so we were basically just not telling them any of the things that we were going through.”—0208, female diagnosed with anal cancer.*



*“[E]ven though I’m in my 40s and my parents are in their 70s and I’m supposed to be taking care of them, it made me feel good that it didn’t matter how old I am, they’ll still there to take care of me, even though they shouldn’t be because I should be taking care of them. You know, even though they have their own ailments that they will worry about yours more than theirs.”—0317, female diagnosed with cervical cancer.*



*“What I had to do with her was tell her ‘You know, I need you to understand fully what is going on with me’ because in the beginning I did keep a line of it from my kids because I didn’t want them to worry about me. When I had my hysterectomy, you had to be off your feet for two weeks while I stayed at a friend’s house and made them think I was out of town. My husband came over there one night but I let the kids think that I wasn’t even at home. So, that’s when I had to tell her [my daughter] everything.”—0307, female diagnosed with cervical cancer.*



*“I was hesitant until I had confirmed things, you know, actual things. And then I didn’t want to worry them [my children], but it was one of those things that you know, you got to let them know what’s going on at a certain point because it’s not something that you can hide.”—0105, male diagnosed with oropharyngeal cancer.*


#### 3.1.3. Discomfort Discussing Anatomic Location of Cancer

An additional subtheme that emerged as a barrier to disclosure centered on participants’ difficulties discussing their diagnoses because of the anatomic location of their cancer. Specifically, participants described embarrassment, perceived stigma, and potential judgement from members of their social support networks.


*“Just the embarrassment, I mean the man I had been dating I won’t even tell him. I’ve not even told him. And the nurse told me, don’t worry about it because she said everyone, even the doctor just now said virtually everyone walking the planet is exposed… Extraordinary anxiety level off the charts, embarrassment, shame. It’s just like HIV, people look down on you. Well, you got it because you know, like anyone deserves the disease.”—0209, male diagnosed with anal cancer.*



*“[Y]ou feel weird and then you’re trying to explain that to her. You know, [Laughter] trying to explain like, ‘Well, what type it is and if you never had it back there then how did you get it?’… I wouldn’t tell everybody that I have cancer cause then they wanna know what type of cancer and then I have to sit down and explain the anal cancer and then they look at you like—that look, a real funny look, you know, and then I’ll have to explain myself and I don’t wanna have to do that.”—0206, female diagnosed with anal cancer.*


One participant described how the anatomic location of anal cancer helped to maintain privacy and limited the need to disclose to others due to minimal observable physical changes. Specifically, this participant reported that she did not want to disclose or discuss her cancer due to its anatomic location, and that (at the same time) the anatomic location helped her hide her diagnosis from those to whom she did not wish to disclose.


*“With breast cancer, you would lose your hair. You have your chemo. So I think, you know, it’s something that outwardly people know that’s going on. I would keep it as quiet as I could until they questioned me on it.”—0211, female diagnosed with anal cancer.*


One participant diagnosed with vulvar cancer reported feeling more comfortable disclosing her diagnosis to other women than with men within her social support networks due to the anatomic location of her cancer.


*“Well, I don’t know that I fear being judged. It’s just uncomfortable to talk about your genitals. (chuckles). I mean, I suspect that men who have testicular cancer have the same problem, you know. Like, it’s difficult in conversation to, to talk about it.”—0306, female diagnosed with vulvar cancer.*


### 3.2. Facilitators of Initial Disclosure

Three unique subthemes were observed among participants which described facilitators of initial disclosure. These reasons included (1) using self-disclosure as a way of seeking support (social and logistic) during cancer treatment; (2) misunderstanding of the link between HPV and cancer in facilitating disclosure to others; and (3) editorializing specifics of a cancer diagnosis to avoid judgment from others.

#### 3.2.1. Disclosure as a Facilitator of Social Support

Participants reported fully disclosing their diagnoses and status of care to individuals within their social networks as a means of facilitating logistic and other forms of support. Some common ways in which loved ones supported participants during their cancer experience included helping them to navigate treatment, providing transportation, and/or making accommodations for participants’ work.


*“I just kept it with my kids. And the only reason I just had was cause I had to have them drive me and pick me up and that kind of stuff.”—0211, female diagnosed with anal cancer.*



*“I let them know what I knew. The important people like my supervisor, my fellow co-workers that I work with. Because the job I do they would have to have somebody replace me as, you know, until I get better. I didn’t announce it to everybody. It was only just a select few that knew what was going on.”—0308, female diagnosed with cervical cancer.*


#### 3.2.2. Misunderstanding of the Link Between HPV and Cancer

An unexpected facilitator of disclosure identified in this study was participants’ misunderstanding of the link between HPV and their cancer diagnosis. Some of the explanations used by participants in disclosing to individuals within their social support networks included topics related to their cancer type being hereditary, HPV not being associated with individual behavior, and the lack of understanding that HPV is an STI. This paradoxical finding aligns with the Disclosure Model, as these misconceptions altered participants’ risk–benefit calculations, reducing anticipated stigma and making disclosure feel safer.


*“And the extended family, you know we’ve talked about the idea that some things can be hereditary and there was a need for follow-up, a conversation I had with my children and my siblings. I have a sister that’s older and a brother that’s younger. And along the way with this—because I—ahead of that getting a colonoscopy we talked about them having the need to do that.”—0202, male diagnosed with anal cancer.*



*“People wanted to know more about it and I told people you know, what I knew but you know, it—it was unlike many other things that you could have where people would know someone who have had it because I didn’t meet anyone who knew someone who had—that had had anything like that. So it was definitely educational to just be able to tell people about it and encourage people to have colonoscopies obviously and just looking at it as a screening thing.”—0208, female diagnosed with anal cancer.*



*“No, I didn’t have a problem talking about it, you know. Cancer is—it’s out there, you know? You never know who’s gonna get cancer or why they got the cancer, you know? Well, yeah, you can with lung cancer, you know, from smoking or whatever, but for—as far as the rectal cancer, it’s hard to say what caused it.”—0210, female diagnosed with anal cancer.*



*“I can’t really say no because in the beginning when people ask me—when they found out and then they ask you what kind it is when you say cervical they automatically—automatically somebody think, ‘Oh, STD’. But that’s not the case because you don’t have to have STDs or anything to get it. It’s from a virus that everybody can get or anybody—even boys can get it. It’s kind of like almost in your body.”—0317, female diagnosed with cervical cancer.*


#### 3.2.3. Editorializing Specifics of Cancer Diagnosis

Another identified facilitator of disclosure was the use of vague terms or intentionally misrepresenting the cancer type when disclosing to others. This strategy was particularly observed among participants diagnosed with anal cancers. These decisions appeared to be motivated by a desire to minimize negative reactions and facilitate social support. Participants with anal cancer often referred to their diagnosis as “rectal” or “colon” cancer instead of explicitly stating “anal” cancer. These language choices highlight how stigma shapes disclosure decisions.


*“I would say something came up in my colonoscopy. Now I have some abnormal cells.”—0208, female diagnosed with anal cancer.*



*“I like to say it’s rectal cancer more than anal cancer just because it sounds better to me. And I usually don’t say anything like that HPV-related or anything like that.”—0204, Male diagnosed with anal cancer.*



*“Well I [would say], ‘I have something in my groin and the doctor says I have cancer cells and I have to do a treatment with radiation and chemo.”—0205, female diagnosed with anal cancer.*



*“I referred to it more as ‘rectal cancer’ because ‘anal cancer’ I think opened up a whole lot of doors… I just didn’t feel comfortable talking about that. And I really can’t event answer why. It felt more personal than saying you have lymphoma then anal cancer. So ‘rectal cancer’ didn’t bother me as much… I think the anal cancer, to be honest with you, I don’t really like that term. It’s so funny but I really didn’t want most people know that I’m gay. But I just didn’t want to get into [it]—and a good majority of my closest friends know about my HIV status. However, some people do not, and so instead of trying to explain stuff, it was easier for me to say ‘rectal’ because most heterosexuals, ‘Oh, I understand that. I knew someone who had that.’ And when you say ‘anal’, they kinda don’t know what to think or say or whatever. So that’s kinda how I dealt with that.”—0207, male diagnosed with anal cancer.*


### 3.3. Experiences After Disclosure

Lastly, subthemes emerged among participants related to their experiences after disclosure. These subthemes centered around stigma, both internalized and perceived, from others with whom the participants shared their diagnosis.

#### 3.3.1. Internalized Stigma

Some participants reported feelings of guilt, shame, internalized stigma, and self-blame for being diagnosed with an HPV-related cancer. Some of these sentiments centered around other aspects of participants’ lives including sexual orientation and individual sexual behaviors. Participants expressed sentiments of self-blame for potentially engaging in behavior that ultimately resulted in being diagnosed with cancer.


*“…even though several doctors and my radiologist said, ‘It’s becoming more common,’ and that you don’t have to be gay though to… You know, human papillomavirus, it can happen to anybody, and anal cancer is one of the biggest things that happens from that disease. You know, from being exposed to it. So it’s not a gay thing. It’s not a gay disease… There’s a little bit of embarrassment involved with that when I was told that. ‘Well, they’re telling me this is from, you know, this and, you know.’ Then, you kinda start going, ‘Well, did my lifestyle kind of maybe have caused that?’ And then I didn’t want to go in that—down that road.”—0207, male diagnosed with anal cancer.*



*“Actually, I was concerned because the—the type of cancer is—was more—it—it can be like a sexually transmitted and I was totally faithful… [Laughter] my whole entire life… [B]ut you know, I couldn’t beat my husband up.”—0211, female diagnosed with anal cancer.*



*“I think as time went on and I was able to process that diagnosis, you know, you feel like, ‘What did I do to deserve this? Did I bring this on?’ This is not my first marriage. In my first marriage, I was abused and I thought, ‘Is there something that, you know? Why did I marry that person? What if he caused it? What if he brought something into my body that was somewhere else?’ So, I guess there’s—because it’s a sexual or a genital cancer, I guess yes, I had some of those of feelings.”—0306, female diagnosed with vulvar cancer.*



*“The main thing I think it was the self-doubt like ‘Could I have caused this? Was there something else I should have been doing?’ You know because HPV was what started me to have the cancer. So, when you say the human papillomavirus, how do I get back when everybody can get it? This is what they tell you when they talk about it. Everybody can get it but it just performs different in people. It lays dormant in some people. Some people comes out very aggressive like it is doing me. But that’s the part that you go ‘Well, what could I have done different? So why is it so aggressive in me?’”—0307, female diagnosed with cervical cancer.*



*Taken further, one participant reported that the feelings of guilt and shame extended beyond the initial diagnosis and subsequently impacted their sexual intimacy. Here, this participant expressed difficulty in engaging in sexual activity after being diagnosed with an HPV-related cancer.*



*“Well, it’s kind of like makes you not want to do things like, you know, sexually active as you were before… [Y]ou just you kind of feel less wanting to do activities like that. You don’t feel the drive as much as before.”–0308, female diagnosed with cervical cancer.*


#### 3.3.2. Perceived Stigma from Others

A total of 13 participants reported general positive experiences with individuals within their social support networks after disclosing their cancer diagnoses. These positive experiences included being openly supported by their close friends and family. Six participants reported neutral experiences after disclosure, while six others reported a mix of both positive and negative experiences. For example, some participants experienced support from their partners but not from their extended family and friends. Other participants experienced the opposite, where their extended family and friends were more supportive than their intimate partners. Finally, two participants reported negative experiences following disclosure. The reasons reported for these negative experiences included perceived blame due to the association of their cancer diagnosis with an STI, limited health literacy among friends and family members, and perceptions about sexual promiscuity due to HPV infection.


*“My daughter didn’t understand me and for me to explain to her as HPV grow and turn into cancer and you get HPV from having intercourse, she pre-judged me because she tried to do her own research. She’s 20 years old. She tried to do her own research and what she came up with HPV was a sexually transmitted disease. In her eyes, I cheated on her father. Now, I have this illness that’s having me in a hospital, having me bed-ridden. So, she had a swordfight with me and another family member told me that that’s what she thought I had a sexually transmitted disease and I was jeopardizing her and—I mean, jeopardize myself or her father while she didn’t read on it correctly. I had to go to her from the other family member that she talked to about it because she was sour with me, because she thought that’s what it was and I got cancer from cheating on her father. That’s what was in her head.”—0307, female diagnosed with cervical cancer.*



*“I didn’t have a lot of sexual partners so it couldn’t have been that. So it’s—I mean so in the beginning you—you are a little bit embarrassed because people automatically think STD.”—0317, female diagnosed with cervical cancer.*



*P: “When I brought that up to him, I didn’t mention the HPV but the HIV, he said, ‘Well, you wouldn’t have all these things if you hadn’t been living the homosexual lifestyle.’ And I told him, I said, ‘Father, you can get HIV no matter who you are.’ He said it’s in the gay community and it’s for a reason he says, ‘Because you’re living an indecent [lifestyle].’*



*I: How do you think that would have played out had you spoken to him about the HPV?*



*P: Same scenario, just he would just be more disgusted with me, and I’ve had others.*



*I: More so than HIV?*



*P: Not more so, he would just be even more disgusted with that like you’re just filled with disease.”—0209, male diagnosed with anal cancer.*


## 4. Discussion

Participants diagnosed with HPV-related cancer provide a unique sample population for better understanding the experiences of individuals diagnosed (often simultaneously) with an STI and cancer. This sample population also provides an opportunity to explore similarities and differences in individual experiences across various cancer types (e.g., oropharyngeal, cervical, anal). Data were collected between 2016 and 2017, during which participants were recruited. At that time, awareness and public discourse around HPV-related cancers were still evolving, and discussions about stigma and disclosure may have differed from more recent perspectives. Despite these contextual differences, the findings remain relevant, as stigma, disclosure concerns, and the need for psychosocial support continue to persist among individuals diagnosed with HPV-related cancers.

This study examined barriers and facilitators to self-disclosure and general communication about HPV-related cancer, along with experiences after disclosure. Some participants immediately shared their cancer diagnoses within their social support networks, while others were more guarded and preferred to only share the information with a limited number of individuals. The main reasons for withholding information about their cancer diagnoses centered around the desire for privacy, minimizing emotional burden on loved ones, and discomfort discussing the anatomic location of their cancer. For participants that were MSM, stigma surrounding both their sexual identity and cancer diagnosis created a ‘double stigma’ that shaped their disclosure decisions. This highlighted how the internal perception of self-blame weighed more heavily in the cancer disclosure decision-making of this subgroup.

Some of the participants that were children of aging parents expressed concerns regarding the frailty of their parents and how disclosure of a cancer diagnosis could negatively impact their well-being. Additionally, participants who were caretakers reported a desire to continue in these roles despite being diagnosed with cancer. This sentiment of managed information disclosure in which individuals diagnosed with cancer strategically coordinate how and when to disclose their diagnoses to their elderly parents is not unique to individuals with HPV-related cancers. That is, many adults diagnosed with cancer report feeling distress related to their dual role as a caregiver and care recipient [39,40]. These findings underscore how this subgroup of participants placed more weight on the anticipated consequences of disclosure among the individuals they were sharing the information with, which to some degree caused role strain in their position within their immediate social network.

Participants also reported a desire to control the flow of information about their cancer diagnosis among individuals within their social networks, including management of who receives this information and who does not. One of the participants diagnosed with anal cancer reported frustrations when individuals within their immediate social network (in this case, her mother) shared the participant’s cancer diagnosis with others. This points to how individuals diagnosed with HPV-related cancers may want to disclose their diagnosis with close family members, but choose not to because of the fear of potential breach of privacy, thus limiting the social support they may receive. These experiences with managing initial disclosure are consistent with previous research in individuals with other forms of cancers [41,42,43]. Applying the Disclosure Model to these findings highlights some of the complex ways that individuals with cancer make decisions regarding initial disclosure of their cancer diagnoses to others. In the context of this study, initial disclosure appears to be decided by each component of Ragins’ model from a combination of internal psychological factors (i.e., being the caretaker, not being taken care of), anticipated negative consequences from those being given information (i.e., disclosing emotionally burdensome news), and perceived additional support from members of a participant’s extended social support network (i.e., assistance in cancer care coordination).

Some participants reported that the lack of outward evidence of being diagnosed with anal cancer (i.e., obvious hair loss, weight loss, or body disfigurement) made it easier to not disclose their conditions to others. Individuals diagnosed with anal cancer frequently reported hesitancy regarding initial disclosure due to difficulties discussing the anatomic site of their cancer. Notably, participants diagnosed with anal cancer experienced positive reactions after disclosure in situations where they specifically editorialized their diagnoses when initially disclosing. While anal cancer may reduce one’s need to disclose (i.e., easier to hide in comparison to oropharyngeal cancer), participants mentioned that hesitancy in disclosing their anal cancer diagnoses stemmed from fear of having to explain how they may have developed this cancer, as well as the possibility of being blamed for their diagnosis due to sexual behavior. The mixed emotions expressed by participants in this study are consistent with the existing literature on experiences of individuals diagnosed with anal cancer [13,24,28]. The “invisibility” of anal cancers has been previously reported as a way of avoiding social stigma from others [25]. Taken together, these findings can be explained conceptually by applying both the Attribution Theory and the Disclosure Model. Here, we observed that participants diagnosed with anal cancer reported not disclosing their diagnoses after weighing out the cost–benefit of potential external stigma, rejection, or blame in exchange for social support for a condition that could be perceived as “self-inflected” because of their conscious behaviors [31].

Another notable finding from this study is that the patients diagnosed with oropharyngeal cancers did not express sentiments of stigma associated with sexual behaviors or self-blame like those diagnosed with anal or gynecological cancers. This contrasts previous studies that have reported on the sentiments of embarrassment experienced by these patients and their limited willingness to discuss the cause of their cancers with others, including the role of oral sex in the transmission of HPV [23]. One possible explanation for this discordance could be related to some of the patients in this study having limited health literacy regarding the link between sexual activity, HPV, and their oropharyngeal cancer diagnosis.

While some participants did share concerns regarding emotionally burdening others with their cancer diagnoses, those that fully disclosed their diagnoses did so in the context of perceiving others as sources of logistic support such as transportation to medical appointments or accommodations at work. The perceived benefit of the logistic support appeared to outweigh the cost of disclosure when applying the Disclosure Model to these cases. Initial disclosure seemed easier among participants with misunderstanding regarding the causes of their HPV-related cancers. Some participants reported sharing their diagnoses with members of their social support networks out of concerns of heritability of these conditions, early detection through screening tests like colonoscopies, and the lack of understanding of HPV as both an STI and a direct cause of cancer. In these cases, self-disclosure may be explained by the mitigation of negative internal psychological factors such as seeing themselves as responsible for their condition. That is, participants with misunderstandings about HPV-related cancers may not hold views or beliefs that their conditions are a direct result of their own actions, which facilitates disclosure when applying Ragins’ model.

One particular strategy employed by participants diagnosed with anal cancers was editorializing their diagnoses when initially disclosing them to others. This included using vague terms or intentionally misrepresenting their cancer type, referring to having “rectal” versus anal cancer. These decisions appeared to be made in minimizing the negative reactions from others in exchange for social support. Similar disclosure behaviors have been previously reported among individuals diagnosed with anal cancer, where terms such as “bowel cancer” [24] or “down there” [25] were used in place of explicitly stating anal cancer. A nuance noted in the responses of male participants diagnosed with anal cancer, who also self-identified as sexual minorities, was the added layer of double or compounded stigma from both belonging to this group and being diagnosed with a cancer directly linked to an STI. This ties back to the internal and external sentiments of blame for a way of living that may be perceived as sexually promiscuous or associated with unhealthy choices [14,28,44]. When analyzed in the context of Ragins’ Disclosure Model, we can see that even in situations where participants may have negative internal attributions of their HPV-related cancer or have negative anticipated consequences in honestly disclosing their condition, editorializing their condition offered a way of seeking social support while avoiding social stigma from others.

### Limitations

Although this study provides unique and nuanced insights on the experiences and disclosure practices of patients with HPV-related cancers, several noteworthy limitations should be acknowledged. First, this study was conducted with a small sample size within one region of West Central Florida, which may limit the transferability of findings. Future work should include a larger, more diverse sample across different geographic regions to examine how regional and cultural differences influence disclosure decisions and communication. Additionally, there may have been some selection bias due to the recruitment process for this study. Second, this study primarily used a qualitative approach in understanding the challenges and psychosocial distress experienced by patients diagnosed with HPV-related cancers. Future work should quantitatively assess whether this patient population experiences clinically significant psychiatric co-morbidities following their cancer diagnoses. While this study focuses on the social and psychological aspects of disclosure, non-disclosure may also have clinical implications, such as delayed care, reduced treatment adherence, and limited patient–provider communication. Future research should explore how disclosure patterns influence medical outcomes.

## 5. Conclusions

The findings from the present study highlight some of the barriers and facilitators faced by individuals with HPV-related cancers during disclosure of their diagnosis and seeking social support. The findings additionally delineate some of the similarities and differences individuals may experience after being diagnosed, particularly when considering the anatomic site of the HPV-related cancer. Thus, it is important for healthcare providers to identify some of these challenges when overseeing the care of this group of cancer patients and assist in linking them with additional social support services after the initial diagnosis. Future research should involve formulating a brief, validated, and multi-dimensional screening measure that would help healthcare providers assess for internalized stigma and limited social support as a way of providing this group of patients additional integrated support services earlier. Additionally, oncology nurses, social workers, and mental health professionals could integrate safe disclosure conversations into routine care to reduce distress and improve support. Public health campaigns and patient education initiatives may further reduce misinformation and normalize conversations about HPV-related cancers. Implementing these strategies within multidisciplinary oncology care could enhance patient well-being, promote informed decision-making, and improve overall quality of life.

## Figures and Tables

**Figure 1 healthcare-13-00966-f001:**
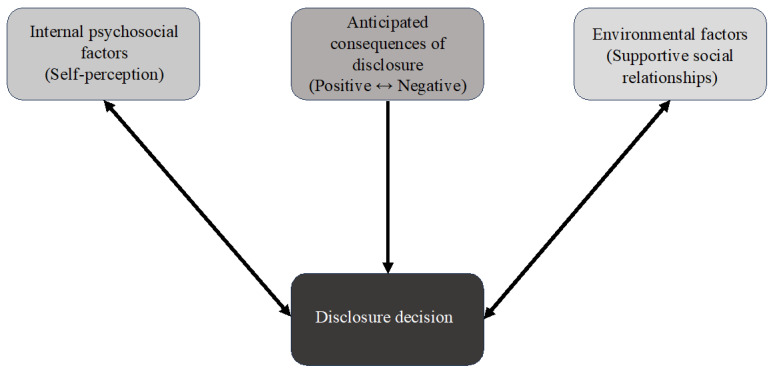
Disclosure Model adapted from Ragins (2008) [32].

**Table 1 healthcare-13-00966-t001:** Demographic characteristics (n = 27) *.

Characteristic	M (SD)	*N* (%)
**Age (Years)**	54.8 (11.5)	
**Gender**		
Female		15 (55.6)
Male		12 (44.4)
**Race/ethnicity**		
White/Caucasian		21 (77.8)
Black/African American		4 (14.8)
Hispanic/Latino		2 (7.4)
**Marital status**		
Married/living with partner		17 (63.0)
Single		3 (11.1)
Divorced		3 (11.1)
Widowed		4 (14.8)
**Sexual orientation**		
Straight/heterosexual		23 (85.2)
Gay/lesbian/homosexual		4 (14.8)
**Education**		
High school/ged/technical		9 (34.6)
Some college		8 (30.8)
Bachelor’s degree or higher		9 (34.6)
**Income (US Dollars)**		
≤$20,000		11 (40.7)
$20,001–40,000		8 (29.6)
>$40,001		8 (29.6)
**Cancer Type**		
Anal		11 (41.0)
Gynecological **		9 (33.0)
Oropharyngeal		7 (26.0)

* Missing data are not included; percentages may not total 100. ** Gynecological cancers include cervical, vulvar, or vaginal sites.

## Data Availability

Data are available on request.

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
