# Peer review of "Understanding Disclosure Decisions and Communication About HPV-Related Cancer: A Qualitative Exploration of Stigma and Social Support"

_healthcare, 2025, doi:10.3390/healthcare13090966_

Round 1
Reviewer 1 Report
Comments and Suggestions for Authors
Title and Abstract
- Your title clearly signals a focus on “disclosure decisions and communication” regarding HPV-related cancer. However, it is worth mentioning the key outcome (stigma/social support) explicitly. For example, “Understanding Disclosure Decisions and Communication about HPV-Related Cancer: A Qualitative Exploration of Stigma and Social Support”
- The abstract is clear and captures main findings. Yet, the final sentence about improved social support resources is brief. Expanding on how or why improved resources might address the revealed barriers will clarify the study’s contribution.
Introduction
- You provide a strong epidemiological backdrop but could strengthen the justification for exploring disclosure decisions. For instance, highlight more explicitly why these decisions matter for psychological outcomes, continuity of care, or partner notification.
- While you introduce stigma well, it might help to consolidate references on how sexual stigma intersects with HPV positivity. This is especially true for individuals with anogenital cancers. Briefly referencing gender/sexual orientation intersectionality up front could better foreshadow your later findings.
- You mention that oropharyngeal cases have surpassed cervical cancer incidence in men, yet you don’t fully align this with why stigma and disclosure remain under-explored for oropharyngeal subpopulations. Articulating that gap in knowledge can boost the paper’s significance.
Methods
- Regarding sample and recruitment, you clearly define inclusion criteria and your focus on oropharyngeal/anogenital tumors. However, specify the approximate clinic volume or patient population from which you recruited. This additional context helps gauge representativeness, even if this is a qualitative study.
- On the rationale for applying Attribution Theory and the Disclosure Model, briefly elaborate on how these frameworks directly informed your codebook or theme development – beyond the mention that you used them for “thematic analysis.” Doing so underscores analytic rigor for your readership.
- Because disclosure is inherently personal, elaborate on how you ensured data security and participant comfort during recruitment and interviews. This strengthens the ethical reporting dimension.
Results
- On your sample characteristics, you note demographic details clearly in Table 1, but the grouping or labeling (e.g., combining all “gynecological” tumors) might miss distinctions among cervical, vulvar, or vaginal sites. A short rationale clarifying these combined categories would help readers interpret the differences in disclosure patterns.
- The paper reveals that misconceptions (e.g., “my cancer is hereditary” or “HPV is not an STI”) ironically enabled some participants to disclose. That is an important but somewhat paradoxical finding. More direct text linking these specific misconceptions to the Disclosure Model or to stigma theory would emphasize the significance.
- This strategy is a central, unique insight. The paper could include a short bridging paragraph explaining why “anal” is uniquely stigmatized. This might contextualize why participants shift to “rectal” or “colon.” Doing so clarifies to readers unfamiliar with HPV stigma how strongly language shapes disclosure.
Discussion
- Regarding the integration of theoretical framework, the Discussion uses Attribution Theory and Disclosure Model fairly well, but it reads at times as if these theories are simply mirrored in participant quotes. Consider stepping back to interpret why certain subgroups (e.g., men who have sex with men, or older caregivers) weigh perceived blame more heavily. Addressing these complexities can deepen your theoretical contribution.
- In comparing your findings with prior literature, your references to existing qualitative studies on anal and oropharyngeal cancer are valuable. You might expand the oropharyngeal dimension: some literature suggests men with HPV-related oropharyngeal cancer face stigma around “oral sex” discussions – is that consistent with or different from your participants’ experiences? Showcasing such parallels or divergences solidifies your study’s place in the broader landscape.
- Regarding study limitations and transferability, you rightly mention the single-institution sample. Possibly highlight how your Southeastern US context (or demographic composition) might shape stigma differently than, say, a more urban or internationally diverse setting. Brief reflection on cultural context can guide future replication or adaptation.
- Clinical Implications
The audience often expects a concluding paragraph linking to practice or policy. For instance: - What might an ideal screening measure for stigma look like?
- How might oncology nurses or social workers incorporate a “safe disclosure conversation” in routine care?
Recommending specific feasible interventions can make your conclusion more actionable.
Conclusion
- The final paragraphs might reinforce a call to implement social support interventions that directly address stigma or misinformation about HPV. Stressing the potential benefits of multidisciplinary care (e.g., mental health professionals, allied health, specialized counseling) could help readers see immediate practical outcomes.
- Future Research
For future research, you hint at wanting larger-scale or multi-region studies. State more concretely that prospective research might examine how specialized communication training for oncology staff can reduce patient reluctance to share. Or how a well-designed educational tool might dispel “hereditary” or “non-STI” myths.
The English could be improved to more clearly express the research.
Author Response
We are very appreciative for the thoughtful feedback provided by the reviewers. We have responded to the reviewers’ comments and believe that the incorporation of their feedback has strengthened this manuscript.
Reviewer 1:
Title and Abstract
Comment: Your title clearly signals a focus on “disclosure decisions and communication” regarding HPV-related cancer. However, it is worth mentioning the key outcome (stigma/social support) explicitly. For example, “Understanding Disclosure Decisions and Communication about HPV-Related Cancer: A Qualitative Exploration of Stigma and Social Support”
Response: We greatly appreciate the reviewer’s suggestion on this and have modified the title to reflect these changes.
Comment: The abstract is clear and captures main findings. Yet, the final sentence about improved social support resources is brief. Expanding on how or why improved resources might address the revealed barriers will clarify the study’s contribution.
Response: We have now clarified the improved social support resources, how they can be implemented, and why they can help address this barrier. This is reflected in the abstract.
Introduction
Comment: You provide a strong epidemiological backdrop but could strengthen the justification for exploring disclosure decisions. For instance, highlight more explicitly why these decisions matter for psychological outcomes, continuity of care, or partner notification.
Response: We have now added a justification for the importance of disclosure decisions into the introduction.
Comment: While you introduce stigma well, it might help to consolidate references on how sexual stigma intersects with HPV positivity. This is especially true for individuals with anogenital cancers. Briefly referencing gender/sexual orientation intersectionality up front could better foreshadow your later findings.
Response: We have briefly addressed the intersection of sexual stigma, gender, and sexual orientation in HPV-related cancers.
Comment: You mention that oropharyngeal cases have surpassed cervical cancer incidence in men, yet you don’t fully align this with why stigma and disclosure remain under-explored for oropharyngeal subpopulations. Articulating that gap in knowledge can boost the paper’s significance.
Response: We have now highlighted why stigma and disclosure remain under-explored for oropharyngeal cancers, whereas cervical and anogenital cancers are more widely recognized in this context.
Methods
Comment: Regarding sample and recruitment, you clearly define inclusion criteria and your focus on oropharyngeal/anogenital tumors. However, specify the approximate clinic volume or patient population from which you recruited. This additional context helps gauge representativeness, even if this is a qualitative study.
Response: Thank you for the feedback. We have incorporated additional specifics to this section of the manuscript.
Comment: On the rationale for applying Attribution Theory and the Disclosure Model, briefly elaborate on how these frameworks directly informed your codebook or theme development – beyond the mention that you used them for “thematic analysis.” Doing so underscores analytic rigor for your readership.
Response: We have elaborated on how Attribution Theory and the Disclosure Model directly informed our codebook and theme development.
Comment: Because disclosure is inherently personal, elaborate on how you ensured data security and participant comfort during recruitment and interviews. This strengthens the ethical reporting dimension.
Response: Thank you for the feedback. We have further expanded this section with more details.
Results
Comment: On your sample characteristics, you note demographic details clearly in Table 1, but the grouping or labeling (e.g., combining all “gynecological” tumors) might miss distinctions among cervical, vulvar, or vaginal sites. A short rationale clarifying these combined categories would help readers interpret the differences in disclosure patterns.
Response: Thank you for the feedback. We included this information in this section. Additionally, we have added this footnote to Table 1.
Comment: The paper reveals that misconceptions (e.g., “my cancer is hereditary” or “HPV is not an STI”) ironically enabled some participants to disclose. That is an important but somewhat paradoxical finding. More direct text linking these specific misconceptions to the Disclosure Model or to stigma theory would emphasize the significance.
Response: Thank you for the feedback. We agree with the reviewer and have further expanded on this section to highlight this finding.
Comment: This strategy is a central, unique insight. The paper could include a short bridging paragraph explaining why “anal” is uniquely stigmatized. This might contextualize why participants shift to “rectal” or “colon.” Doing so clarifies to readers unfamiliar with HPV stigma how strongly language shapes disclosure.
Response: Thank you for the feedback. We agree with the reviewer and have further expanded on this section to clarify this unique finding.
Discussion
Comment: Regarding the integration of theoretical framework, the Discussion uses Attribution Theory and Disclosure Model fairly well, but it reads at times as if these theories are simply mirrored in participant quotes. Consider stepping back to interpret why certain subgroups (e.g., have sex with men, or older caregivers) weigh perceived blame more heavily. Addressing these complexities can deepen your theoretical contribution.
Response: Thank you for the insightful feedback. We have added language in this section to deepen the theoretical connection.
Comment: In comparing your findings with prior literature, your references to existing qualitative studies on anal and oropharyngeal cancer are valuable. You might expand the oropharyngeal dimension: some literature suggests men with HPV-related oropharyngeal cancer face stigma around “oral sex” discussions – is that consistent with or different from your participants’ experiences? Showcasing such parallels or divergences solidifies your study’s place in the broader landscape.
Response: We appreciate this insightful feedback. We have added some additional text to the discussion articulating some of the differences between our findings among patients with oropharyngeal cancers and prior work.
Comment: Regarding study limitations and transferability, you rightly mention the single-institution sample. Possibly highlight how your Southeastern US context (or demographic composition) might shape stigma differently than, say, a more urban or internationally diverse setting. Brief reflection on cultural context can guide future replication or adaptation.
Response: Thank you for the feedback. We have modified this section to reflect these changes.
Clinical Implications
Comment: The audience often expects a concluding paragraph linking to practice or policy. For instance:
- What might an ideal screening measure for stigma look like?
- How might oncology nurses or social workers incorporate a “safe disclosure conversation” in routine care?
Recommending specific feasible interventions can make your conclusion more actionable.
Response: Thank you for the insightful feedback. We have modified this section to reflect these changes.
Conclusion
Comment: The final paragraphs might reinforce a call to implement social support interventions that directly address stigma or misinformation about HPV. Stressing the potential benefits of multidisciplinary care (e.g., mental health professionals, allied health, specialized counseling) could help readers see immediate practical outcomes.
Response: We have expanded the limitations section to reflect these changes.
Future Research
Comment: For future research, you hint at wanting larger-scale or multi-region studies. State more concretely that prospective research might examine how specialized communication training for oncology staff can reduce patient reluctance to share. Or how a well-designed educational tool might dispel “hereditary” or “non-STI” myths.
Response: We have expanded this section to provide more concrete future research directions. Specifically, we now emphasize the need for specialized communication training for oncology staff to help reduce patient reluctance to disclose their diagnosis. Additionally, we highlight the potential role of educational tools in dispelling misconceptions about HPV-related cancers, particularly myths regarding hereditary transmission and HPV as a non-STI. These additions ensure that future research recommendations are more actionable and aligned with practical applications in oncology care.
Reviewer 2 Report
Comments and Suggestions for Authors
Dear Authors,
congratulation for the research you conducted and the resulting research paper. Overall, I recommend to Editor/s a "minor revision" since I think that the paper just needs minor changes and enhancements.
- Consider adding this reference along with reference n. 3, which is more updated and thus would enhance your position within the literature:
Testoni, I.; Nicoletti, A.E.; Moscato, M.; De Vincenzo, C. A Qualitative Analysis of the Experiences of Young Patients and Caregivers Confronting Pediatric and Adolescent Oncology Diagnosis. Int. J. Environ. Res. Public Health 2023, 20, 6327. https://doi.org/10.3390/ijerph20146327
-
From a methodological perspective, while the qualitative approach allows for rich and nuanced insights, there seems to be a lack of clarity in describing its research design. Several aspects could be improved to enhance transparency and rigor:
- Unclear participant selection process: You state that participants were identified through a hospital electronic health record system (lines 178-179), but you do not provide detailed exclusion criteria. It is unclear whether any patients were excluded and on what basis, which raises concerns about selection bias. Additionally, participants were recruited from a hospital setting, meaning they were already engaged with the healthcare system. This may exclude individuals who avoid medical care due to fear of stigma, potentially underestimating the challenges some patients face.
- Limited explanation of data analysis: You mention using MAXQDA (lines 212-213) for qualitative coding but more details are need to describe the coding strategy. A more structured explanation of the thematic analysis process, including how codes were developed and refined, would strengthen the methodological rigor.
-
Aside from these methodological concerns, there are additional areas where the study could be improved:
- Data collection: please add some reflections about the time in which you collected data (2016/2017) as some processes might have been changed since than.
- Interpretation not always supported by data – You claim that patients with oropharyngeal cancer were more open to disclosure compared to those with anal cancer, attributing this difference to the visibility of the disease. However, the study lacks quantitative data to robustly support this statement, and the discussion section does not address this issue. It could be contextualized and further explored in the discussion section.
- Limited discussion of clinical consequences: The study focuses primarily on social and psychological aspects but does not explore how non-disclosure might affect medical outcomes, such as treatment adherence or timely access to care. Expanding this discussion would add valuable context.
Author Response
We are very appreciative for the thoughtful feedback provided by the reviewers. We have responded to the reviewers’ comments and believe that the incorporation of their feedback has strengthened this manuscript.
Reviewer 2:
Dear Authors,
congratulation for the research you conducted and the resulting research paper. Overall, I recommend to Editor/s a "minor revision" since I think that the paper just needs minor changes and enhancements.
Comment: Consider adding this reference along with reference n. 3, which is more updated and thus would enhance your position within the literature:
Testoni, I.; Nicoletti, A.E.; Moscato, M.; De Vincenzo, C. A Qualitative Analysis of the Experiences of Young Patients and Caregivers Confronting Pediatric and Adolescent Oncology Diagnosis. Int. J. Environ. Res. Public Health 2023, 20, 6327. https://doi.org/10.3390/ijerph20146327
Response: We appreciate your suggestion and have added an updated reference to support our revisions.
From a methodological perspective, while the qualitative approach allows for rich and nuanced insights, there seems to be a lack of clarity in describing its research design. Several aspects could be improved to enhance transparency and rigor:
Comment: Unclear participant selection process: You state that participants were identified through a hospital electronic health record system (lines 178-179), but you do not provide detailed exclusion criteria. It is unclear whether any patients were excluded and on what basis, which raises concerns about selection bias. Additionally, participants were recruited from a hospital setting, meaning they were already engaged with the healthcare system. This may exclude individuals who avoid medical care due to fear of stigma, potentially underestimating the challenges some patients face.
Response: Thank you for the thoughtful feedback. We have modified the methods section of the manuscript to further detail the participant selection process. We also added an acknowledgement of the possible selection bias in the limitations section. While the reviewer’s point regarding the possible exclusion of patients who are not engaged with the healthcare system is well taken, we think this may not be relevant to the present study since we primarily had to identify patients with confirmed HPV-associated cancers in order to recruit them to the study. It would be difficult to recruit patients not engaged with the healthcare system into the study as we would not have a way of confirming their diagnosis in the first place.
Comment: Limited explanation of data analysis: You mention using MAXQDA (lines 212-213) for qualitative coding but more details are need to describe the coding strategy. A more structured explanation of the thematic analysis process, including how codes were developed and refined, would strengthen the methodological rigor.
Response: Thank you for the thoughtful feedback. We have expanded on the methods section of the manuscript to include more details on the coding strategy.
Aside from these methodological concerns, there are additional areas where the study could be improved:
Comment: Data collection: please add some reflections about the time in which you collected data (2016/2017) as some processes might have been changed since than.
Response: We have added reflection on the data collection period in discussion section:
Comment: Interpretation not always supported by data – You claim that patients with oropharyngeal cancer were more open to disclosure compared to those with anal cancer, attributing this difference to the visibility of the disease. However, the study lacks quantitative data to robustly support this statement, and the discussion section does not address this issue. It could be contextualized and further explored in the discussion section.
Response: Thank you for the thoughtful feedback. We agree with the reviewer and have removed this language from the discussion.
Comment: Limited discussion of clinical consequences: The study focuses primarily on social and psychological aspects but does not explore how non-disclosure might affect medical outcomes, such as treatment adherence or timely access to care. Expanding this discussion would add valuable context.
Response: We have incorporated this discussion into the Limitations section.
Reviewer 3 Report
Comments and Suggestions for Authors
Dear authors,
I enjoyed reading your paper. I think that this paper is well structured, done and written.
The introduction provides enough information about the topic. The methodology and results are appropriate and well-presented. The discussion is well written, but the study has limitations mentioned at the end (small sample size, etc.).
Did the level of education impact how individuals in this study self-disclosed? If possible, please elaborate.
I must say that I am not a public health or social medicine specialist but a clinical microbiologist. I enjoyed reading this kind of paper, and it helped me see the problem of HPV-related cancers from a different perspective (patient point of view).
Author Response
Reviewer 3:
Dear authors,
I enjoyed reading your paper. I think that this paper is well structured, done and written.
The introduction provides enough information about the topic. The methodology and results are appropriate and well-presented. The discussion is well written, but the study has limitations mentioned at the end (small sample size, etc.).
Did the level of education impact how individuals in this study self-disclosed? If possible, please elaborate.
I must say that I am not a public health or social medicine specialist but a clinical microbiologist. I enjoyed reading this kind of paper, and it helped me see the problem of HPV-related cancers from a different perspective (patient point of view).
Response: We would like to thank the reviewer for their feedback. We did not observe notable differences by education level. This could be due to the small sample size of the study. Future work with larger sample sizes may be able to specifically assess if education plays a role in self-disclosure.
Round 2
Reviewer 1 Report
Comments and Suggestions for Authors
- While the introduction is more comprehensive than the initial version, clarifying how oropharyngeal cancer incidence surpassing cervical cancer in men ties into the study’s rationale would strengthen its cohesiveness. Also highlight psychosocial risks for subgroups (like MSM) to further anchor your research objectives.
- The overall design is appropriate for qualitative inquiry; however, a clearer explanation of how theoretical frameworks (Attribution Theory and Disclosure Model) specifically influenced data saturation decisions or final sample size would improve methodological transparency.
- The rationale for combining certain gynecological sites is now more explicit. Nonetheless, elaborating on the timing of data collection (2016–2017) and how that context may shape findings should be briefly emphasized. Also, provide details on how you addressed any emotional distress that emerged during interviews would strengthen the
- The results section is more thorough post-revision but consider a short summary paragraph bridging each major theme to your theoretical framework. This would maintain clarity for readers less familiar with your coding approach, especially in the subthemes on “editorializing the diagnosis.”
- The conclusion is now better and stronger. A tighter recap of key insights, for example, that oropharyngeal cases in your sample showed fewer direct concerns about sexual stigma—would help differentiate your findings from prior research more explicitly. Also, clarifying how these insights might guide immediate clinical practices (e.g., short screening measures for stigma in oncology clinics) would increase real-world applicability.
Comments on the Quality of English Language
Overall clarity is decent, but certain sentences (especially in the discussion) can be condensed or reorganized for smoother flow. Minor grammar edits and consistent usage of terms (e.g., “oropharyngeal vs. orophar.,” “anal vs. anogenital,” etc.) would improve readability.
Author Response
We are very appreciative of the continued feedback on improving the manuscript. Below are our responses to the reviewer comments:
- While the introduction is more comprehensive than the initial version, clarifying how oropharyngeal cancer incidence surpassing cervical cancer in men ties into the study’s rationale would strengthen its cohesiveness. Also highlight psychosocial risks for subgroups (like MSM) to further anchor your research objectives.
- Response: Thank you for the feedback. We included some of this language in the Introduction section on the previously submitted edits. We added additional language regarding the psychosocial risks among subgroups like MSM.
- The overall design is appropriate for qualitative inquiry; however, a clearer explanation of how theoretical frameworks (Attribution Theory and Disclosure Model) specifically influenced data saturation decisions or final sample size would improve methodological transparency.
- Response: Thank you for the feedback. We added additional language to the methods section per the recommendations of the reviewer.
- The rationale for combining certain gynecological sites is now more explicit. Nonetheless, elaborating on the timing of data collection (2016–2017) and how that context may shape findings should be briefly emphasized. Also, provide details on how you addressed any emotional distress that emerged during interviews would strengthen the
- Response: Thank you for the response. We included some of this language in the Discussion section on the previous edits submitted. We additionally included language regarding handling emotional distress during interviews in the Methods section per the recommendations of the reviewer.
- The results section is more thorough post-revision but consider a short summary paragraph bridging each major theme to your theoretical framework. This would maintain clarity for readers less familiar with your coding approach, especially in the subthemes on “editorializing the diagnosis.”
- Response: Thank you for the feedback. We included some of this language in the previous edits submitted, particularly to this subtheme. We believe that the Discussion section integrates how the themes in the participant responses relate to the theoretical frameworks. We wanted to additionally be mindful of the overall length of the manuscript and for this reason believe that including this language may not add additional value for the reader.
- The conclusion is now better and stronger. A tighter recap of key insights, for example, that oropharyngeal cases in your sample showed fewer direct concerns about sexual stigma—would help differentiate your findings from prior research more explicitly. Also, clarifying how these insights might guide immediate clinical practices (e.g., short screening measures for stigma in oncology clinics) would increase real-world applicability.
- Response: Thank you for the feedback. We included some of this language in the previous edits submitted, specifically relating to clinical practice. Per the recommendations of the other reviewers, we previously modified the manuscript to report the differences in sexual stigma reported by cancer type. Although two of the seven patients with oropharyngeal cancers reported hesitancy in initial disclosure, there was some concern with interpreting this to mean that these patients were less hesitant than those with anal or cervical cancers.
- Comments on the Quality of English Language: Overall clarity is decent, but certain sentences (especially in the discussion) can be condensed or reorganized for smoother flow. Minor grammar edits and consistent usage of terms (e.g., “oropharyngeal vs. orophar.,” “anal vs. anogenital,” etc.) would improve readability.
- Response: Thank you for the feedback. We went through the manuscript and made these changes by removing the term anogenital.